# Structural basis of bile salt extrusion and small-molecule inhibition in human BSEP

Hongtao Liu [1,3], Rossitza N. Irobalieva [1,3], Julia Kowal [1], Dongchun Ni [2], Kamil Nosol [1], Rose Bang-Sørensen [1], Loïck Lancien[1], Henning Stahlberg [2], Bruno Stieger [1] & Kaspar P. Locher [1] ✉

BSEP (ABCB11) is an ATP-binding cassette transporter that is expressed in hepatocytes and extrudes bile salts into the canaliculi of the liver. BSEP dysfunction, caused by mutations or induced by drugs, is frequently associated with severe cholestatic liver disease. We report the cryo-EM structure of glibenclamide-bound human BSEP in nanodiscs, revealing the basis of small-molecule inhibition. Glibenclamide binds the apex of a central binding pocket between the transmembrane domains, preventing BSEP from undergoing conformational changes, and thus rationalizing the reduced uptake of bile salts. We further report two high-resolution structures of BSEP trapped in distinct nucleotide-bound states by using a catalytically inactivated BSEP variant (BSEP$_{E1244Q}$) to visualize a pre-hydrolysis state, and wild-type BSEP trapped by vanadate to visualize a post-hydrolysis state. Our studies provide structural and functional insight into the mechanism of bile salt extrusion and into small-molecule inhibition of BSEP, which may rationalize drug-induced liver toxicity.

Bile salts play a crucial role in the digestion and absorption of dietary fats in the small intestine, particularly in the duodenum[1]. The human bile salt export pump (BSEP), also known as ABCB11, is one of four transporters involved in the enterohepatic circulation of bile salts[2–4]. BSEP, located in the apical membrane of hepatocytes, transports a variety of bile salts into the canaliculi against a steep concentration gradient[5–7]. Its function is closely linked to that of two other hepatobiliary ABC transporters also located in the canalicular membrane—the phosphatidylcholine (PC) translocator ABCB4 and the heterodimeric cholesterol transporter ABCG5/G8[8,9]. The proper function of BSEP, ABCB4, and ABCG5/G8 is essential for bile formation as they mediate the efflux of major bile components (bile salts, PC, and cholesterol, respectively) which form mixed micelles in the bile (Supplementary Fig. 1a).

High intracellular concentrations of bile salts are toxic, causing the disruption of membranes and the accumulation of reactive oxygen species in hepatocytes[10,11]. Mutations in the BSEP gene can impair BSEP activity, leading to the accumulation of bile salts in hepatocytes and ultimately to hepatocyte necrosis or apoptosis in patients[12,13].

Mutation-induced BSEP dysfunction has been associated with a continuous spectrum from mild benign recurrent intrahepatic cholestasis type 2 (BRIC2), to severe, progressive familial intrahepatic cholestasis type 2 (PFIC2)[14,15]. In addition to genetic mutations, BSEP function can also be inhibited by commonly used drugs, which can lead to drug-induced liver injury (DILI) that can progress to cholestatic and mixed patterns of liver damage[5,16]. One such drug is glibenclamide, also known as glyburide, which is used to treat type 2 diabetes[17]. It has been shown to be a non-specific, competitive inhibitor of BSEP[18,19], and glibenclamide-mediated inhibition of BSEP may therefore cause DILI in susceptible patients[5].

BSEP has two transmembrane domains (TMDs), each composed of six TM helices, and two nucleotide-binding domains (NBDs) (Fig. 1a). The NBDs of BSEP are predicted to form one functional nucleotide-binding site (NBS2, consisting of Walker-A and Walker-B motifs of NBD2 and the signature motif of NBD1), and one degenerate nucleotide-binding site (NBS1, consisting of Walker-A and Walker-B motifs of NBD1 and the signature motif of NBD2)[20,21]. To date, one substrate-free (apo) and two substrate-bound structures of BSEP have

[1]Institute of Molecular Biology and Biophysics, ETH Zürich, Zürich, Switzerland. [2]Laboratory of Biological Electron Microscopy, Institute of Physics, School of Basic Science, EPFL, and Department of Fundamental Microbiology, Faculty of Biology and Medicine, University of Lausanne, Lausanne, Switzerland. [3]These authors contributed equally: Hongtao Liu, Rossitza N. Irobalieva. ✉e-mail: locher@mol.biol.ethz.ch

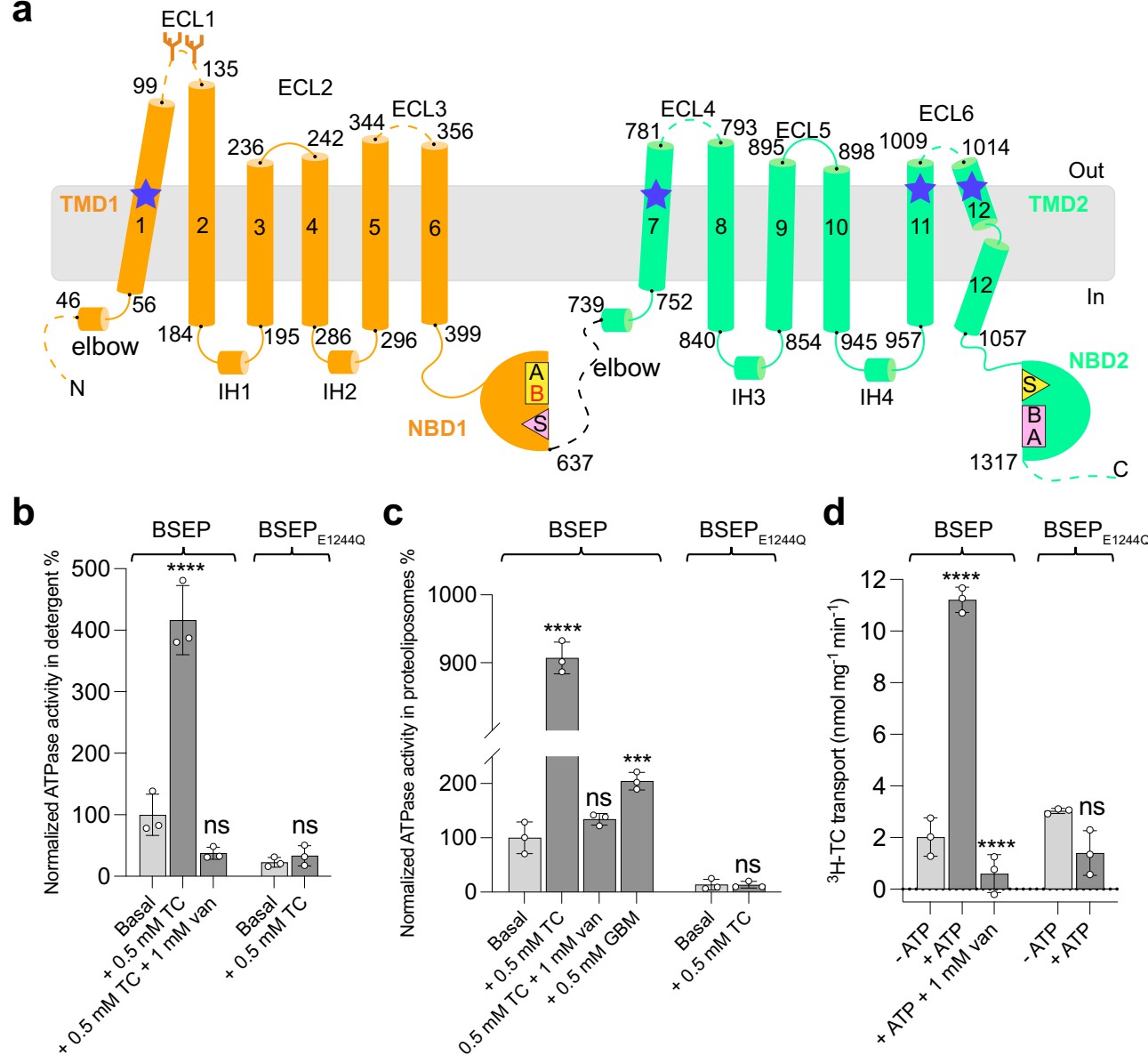

**Fig. 1 | Topology and functional characterization of human BSEP. a** Topology diagram of human BSEP. TM helices (1–12), elbow helices, intracellular helices (IH1–IH4), and extracellular loops (ECL1–ECL6) are numbered and indicated. Stretches of amino acids not resolved in the structures are shown as dashed lines. NBS1 and NBS2 are colored yellow and pink, and the conserved motifs are indicated (A: Walker-A; B: Walker-B; S: Signature motif). The degenerate catalytic site is located in Walker-B labeled in red (B) of NBD1. Blue stars indicate regions that are in contact with GBM. **b, c** Normalized ATPase activity of BSEP and BSEP$_{E1244Q}$ in detergent and proteoliposomes in the presence of substrate TC, GBM, and/or vanadate. **d** ATP-driven transport of $^3$H-TC by BSEP and BSEP$_{E1244Q}$ in proteoliposomes. The observed TC transport rate of BSEP$_{E1244Q}$ in the absence of ATP is considered background. 50.5 µM TC was added and incubated prior to the reaction. The reaction was started by adding ATP and the concentration of ATP was 5 mM throughout in (**b**–**d**). The data points above indicate means of three independent measurements and error bars indicate SD. Statistical significance was determined using ordinary one-way ANOVA, Tukey's multiple comparison test. Differences between basal activity and stimulated activity or between transport activities in the presence or absence of ATP were depicted as ****$P \leq 0.0001$ and ***$P = 0.0001$.

been reported[22,23]. The substrate-free structure of BSEP revealed an inward-facing conformation with a helix (hereafter referred to as the auto-inhibitory helix) formed by the loop connecting NBD1 and TMD2 lodged in the central cavity. The structures of substrate-bound BSEP also adopted an inward-facing conformation, with the central binding pocket occupied by either a single sodium taurocholate (TC) molecule or two TC molecules. While these structures showed well-resolved TMDs, the density of NBDs was too weak to characterize the nucleotide-binding sites in detail.

Because of its clinical relevance, the structure and transport mechanism of BSEP are intensively studied. It is currently unknown

how the small molecules can inhibit BSEP function and how bile salts are extruded and released during the transport cycle. Furthermore, previous structures of BSEP were determined in a detergent solution, which is known to modulate protein function[24]. Here, we report three cryo-electron microscopy (cryo-EM) structures of human BSEP in lipidic nanodiscs. We determine a structure of BSEP bound to the inhibitory compound glibenclamide (GBM), providing insight into the small-molecule inhibition. We further present two structures of BSEP bound to ATP, one in a pre-hydrolysis state and another in a post-hydrolysis state. These structures provide experimental visualization of the functionally important features of the nucleotide-binding sites.

In combination with functional data, our results advance our molecular understanding of the function of BSEP.

## Results

### Functional characterization of human BSEP

Human wild-type BSEP and a variant in which the catalytic glutamate of NBS2 was replaced by a glutamine (E1244Q) were expressed in HEK293 stable cell lines and purified in detergent[25]. In the presence of the substrate TC, the ATPase activity of detergent-purified BSEP increased ~3-fold relative to the basal level (Fig. 1b). The ATPase rate also increased as a function of ATP concentration and TC concentration (Supplementary Fig. 1b, c). We determined an apparent $K_{m,ATP}$ of 207 μM, and an apparent $K_{m, TC}$ of 109 μM for wild-type BSEP (Supplementary Fig. 1b, c). We then reconstituted BSEP into proteoliposomes and observed that the TC-stimulated ATPase activity of BSEP was ~8-fold higher than the basal activity (Fig. 1c). Our results confirm the functional relevance of the lipidic environment for BSEP activity, which caused us to use nanodisc-reconstituted BSEP for structural studies. We also observed that GBM can stimulate the ATPase activity of proteoliposome-reconstituted BSEP with a ~2-fold increase relative to the basal activity (Fig. 1c). No ATPase activity was observed for $BSEP_{E1244Q}$ in detergent or in proteoliposomes (Fig. 1b, c), even at elevated concentrations of ATP and TC (Supplementary Fig. 1b, c). We observed that $^3$H-taurocholate ($^3$H-TC) transport mediated by BSEP in proteoliposomes was strictly dependent on ATP (Supplementary Fig. 1d, e). This was further confirmed by the absence of TC transport by $BSEP_{E1244Q}$ even in the presence of ATP (Fig. 1d). The in vitro transport rate at ~50 μM TC was 11 nmol (TC) mg$^{-1}$ min$^{-1}$, resulting in a turnover number of ~1.6 molecules TC per molecule BSEP per minute (Fig. 1d). We also used vanadate to inhibit BSEP function, by trapping the protein in a post-hydrolysis state[26]. TC-stimulated ATPase activity was inhibited by vanadate both in detergent and in proteoliposomes, and $^3$H-TC transport by proteoliposome-reconstituted BSEP was fully abolished (Fig. 1b–d).

### Structure of glibenclamide-bound BSEP

In the presence of GBM, TC-stimulated ATPase activity of detergent-purified BSEP was decreased, but was not completely abolished (Fig. 2a). $^3$H-TC uptake by proteoliposome-reconstituted BSEP was also reduced by adding GBM in a dose-dependent manner, with an inhibition constant IC$_{50}$ of ~10 μM (Fig. 2b and Supplementary Fig. 2). This result was consistent with previously reported IC$_{50}$ values for GBM inhibition[27].

To gain structural insight into the interaction with GBM, we prepared nanodisc-reconstituted BSEP bound to GBM and determined a cryo-EM structure at 3.2 Å resolution (Supplementary Figs. 3a and 4a). GBM-bound BSEP adopts an inward-facing conformation (Fig. 2d). A single GBM molecule binds within a pocket formed by transmembrane (TM) helices TM1, TM7, and TM12, and GBM adopts a U-shape at the apex of the central cavity of BSEP (Fig. 2c, d). The EM density map suggests that GBM might adopt two distinct orientations when bound to BSEP, which is consistent with the fact that at a first approximation, GBM has pseudo-2-fold molecular symmetry (Fig. 2d and Supplementary Fig. 5). In one of these, the benzamide group extends towards TM7 and TM12, while the sulfonylurea group interacts with TM1 and TM2 (Fig. 2d, top panel). To accommodate this binding pose, residue F1018 is shifted to avoid clashing with the benzamide group of GBM. Alternatively, GBM can be fitted into the density by a 180° rotation relative to the first orientation (Fig. 2d, bottom panel). In this case, the sulfonylurea group interacts with residue F1018, allowing a downward shift of the phenyl group of F1018.

We noted that GBM binds deeper in the central pocket (closer to the external membrane boundary) than TC does (Fig. 2d, e). Thus, GBM promotes an unproductive, inward-facing conformation, thereby interfering with conformational changes required for substrate extrusion, but not with those associated with ATP hydrolysis, as reflected by the stimulation of the ATPase activity in the presence of GBM (Figs. 1c and 2a). It was previously noted that in the substrate-free state[22], TM4 of BSEP adopts a straight conformation, whereas TM4 is kinked in the TC-bound structure[23], resulting in an occluded TC-binding pocket (Supplementary Fig. 6b). GBM-bound BSEP shows an incomplete occlusion of the binding pocket, with TM4 remaining in a straight conformation (Supplementary Fig. 6c).

The binding site of GBM is formed by BSEP residues Q76, L80, F83, F776, N996, and S1022 (Fig. 2d and Supplementary Fig. 6a). To confirm the interactions of these residues with GBM, we mutated them individually and analyzed the resulting ATPase activity. The four mutants Q76A, L80F, F83A, and F776A show a reduction in the GBM-stimulated ATPase activity in comparison to that of wild-type BSEP (Fig. 2f). The two mutants N996A and S1022F exhibit increased basal ATPase activity compared to the wild-type BSEP, and their activity is stimulated by GBM (Fig. 2f). The basis for the increased basal ATPase rates is unclear. The finding that the mutation N996A does not abolish the stimulation suggests that the acetamide group, while in the vicinity of bound GBM, does not strongly contribute to its binding. Given the location of S1022 in the GBM binding pocket, one might expect that the mutation S1022F would generate a clash with bound GBM. However, this is not the case given that the S1022F still displays stimulation of ATPase activity. We conclude that GBM binds the $BSEP_{S1022F}$ in only one of the two possible orientations discussed above, avoiding a clash with the phenyl side chain of F1022 and maintaining GBM-stimulated ATPase activity.

### Structures of nucleotide-bound BSEP

To capture BSEP in a nucleotide-bound state, nanodisc-reconstituted $BSEP_{E1244Q}$ in the presence of TC and ATP was prepared for cryo-EM studies (Supplementary Fig. 3b). Approximately 60% of the particles adopted a closed state and yielded a cryo-EM density map at a resolution of 3.0 Å (Fig. 3a and Supplementary Fig. 7). Strong and well-defined EM densities are observed for both NBDs and TMDs (Supplementary Fig. 8a, b). We observed tightly packed TMDs and NBDs in a closed conformation (Fig. 3a, b). No substrate binding cavity is present between the TMDs, suggesting that this conformation may represent a state of post-substrate release. All the conserved motifs of the transporter are well resolved, including the A-loop, Q-loop, Walker-A, Walker-B, and signature motifs (Fig. 3b–d). We observed two distinct EM densities in the two NBSs, each corresponding to an ATP molecule, which is consistent with the characteristic architecture of other ABC transporters (Fig. 3c, d). ATP binding is mediated by the Walker-A motif interacting with the phosphate groups and the A-loop interacting with the purine ring of ATP through π–π stacking (Fig. 3c–f). We also observed clear densities for bound Mg$^{2+}$ ions in the two NBSs, which are coordinated by the beta and gamma phosphates of ATP, the glutamine residues of the Q-loop (Q503 and Q1161), the serine residues of the Walker-A motifs (S462 and S1120), the aspartate residues of the Walker-B motifs (D583 and D1243), and water molecules.

We next determined the structure of vanadate-trapped BSEP in nanodiscs at a resolution of 2.8 Å (Fig. 4a and Supplementary Fig. 9). The structure reveals a similar overall conformation to that of $BSEP_{E1244Q}$, with a root-mean-square deviation (RMSD) of 0.426 Å between the two structures (Supplementary Table 2). Two strong EM densities for nucleotides are found in the NBSs (Fig. 4c, d and Supplementary Fig. 8c, d). We modeled an ATP molecule into the density at the degenerate NBS1 and ADP-VO$_4^{3-}$ at that of NBS2 (Supplementary Fig. 10). In principle, an ATP molecule could also be bound at NBS2, but we observed that the γ-phosphate of ATP did not fit well into the EM density (Supplementary Fig. 10c). Vanadate replaces the position of the gamma-phosphate of ATP after ATP hydrolysis and mediates the interaction between the catalytic glutamate (E1244) and the attacking water, trapping ADP and preventing its release from NBS2 (Fig. 4d).

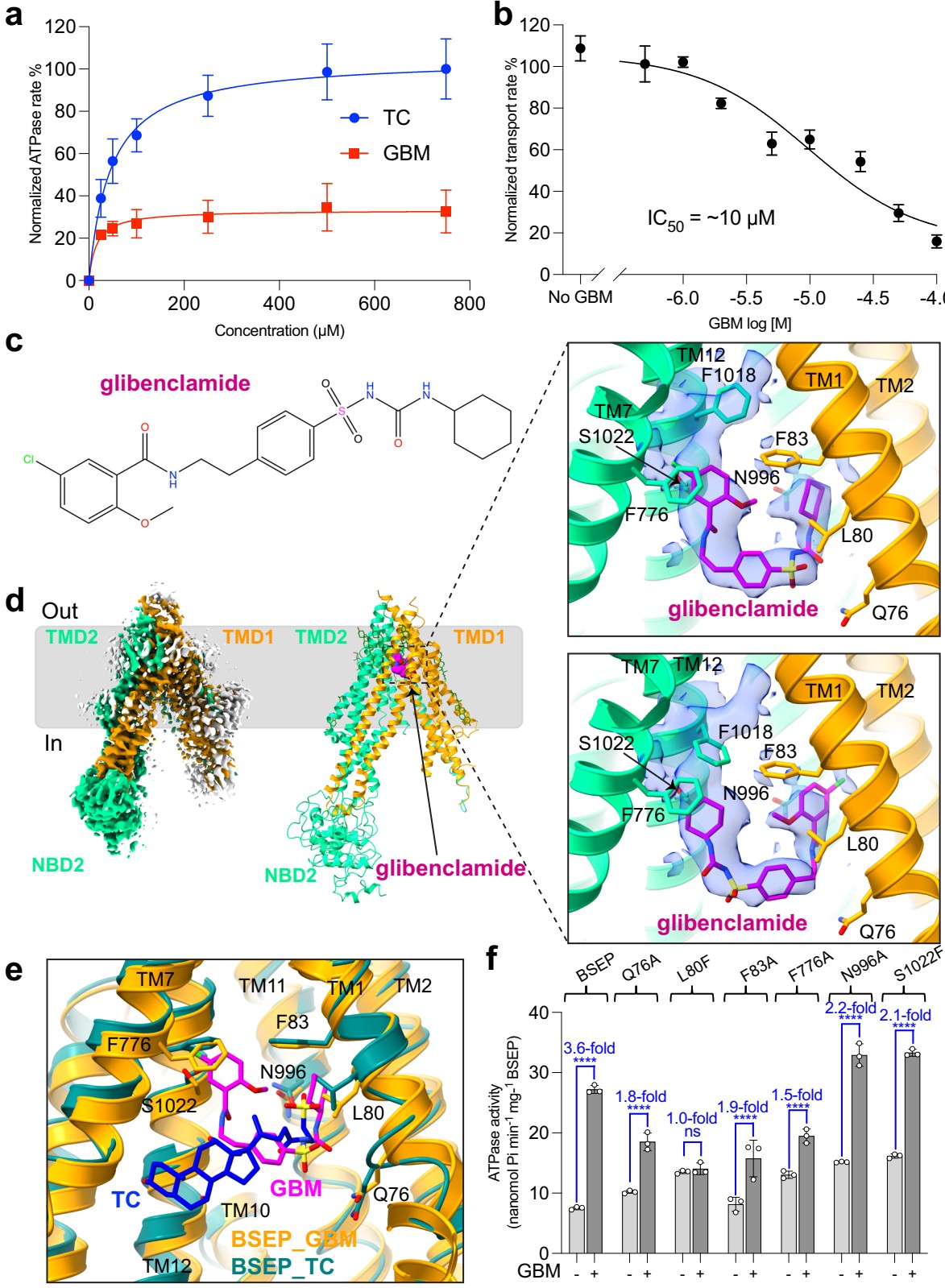

The structures of $BSEP_{E1244Q}$ and vanadate-trapped BSEP reveal a conformation of the TMDs with a closed (collapsed) translocation pathway, with no space for a substrate or inhibitor. In the context of a transport cycle, these structures therefore likely reflect states after substrate release but prior to the release of hydrolyzed nucleotides from the NBDs (Figs. 3a and 4a). $BSEP_{E1244Q}$ represents a state prior to ATP hydrolysis, whereas vanadate-trapped BSEP represents a

state immediately after ATP hydrolysis. While these structures share similarities, the helix connecting the D-loop in NBS1 of vanadate-trapped BSEP is shifted relative to that of $BSEP_{E1244Q}$, resulting in a rearrangement of H615 and D1250 (Supplementary Fig. 11b). Additionally, two residues (E1244 and H1275) at NBS2 are shifted toward the nucleotide in the structure of vanadate-trapped BSEP (Supplementary Fig. 11c). Our structural findings are in line with the

**Fig. 2 | Cryo-EM structure and functional analysis of glibenclamide-bound BSEP. a** Normalized ATPase activity of BSEP in detergent as a function of increasing concentration of TC or GBM. **b** Normalized inhibition of TC transport as a function of GBM concentration in the presence of 50.5 µM TC and 5 mM ATP. The points were plotted using nonlinear regression of the Michaelis-Menten equation and the IC$_{50}$ was calculated to be ~10 µM and the 95% confidence intervals (CI) range is 5–22 µM. **c** Chemical structure of GBM drawn in ChemDraw. **d** Cryo-EM density map and structure of BSEP in a complex with GBM (magenta). The right panels show GBM in two different orientations and the molecular interactions in the central cavity, shown together with the density of GBM (blue). Residues interacting with GBM (Q76, L80, F83, F776, N996 and S1022) are shown as sticks. **e** Comparison of GBM (this study) and TC binding sites (PDB 7E1A). **f** ATPase activity of BSEP and mutants (Q76A, L80F, F83A, F776A, N996A, and S1022F) in the presence or the absence of GBM. The increase of GBM-stimulated ATPase activity is indicated. 300 µM GBM was added and pre-incubated prior to the reaction. 5 mM ATP was used for initiating the assay. Data points in **a**, **b**, and **f** indicate the mean of three independent measurements and error bars indicate SD. Statistical significance in **f** was determined using ordinary one-way ANOVA, Tukey's multiple comparison test. Differences between wild-type BSEP and mutants were depicted as ****$P \leq 0.0001$.

functional attributes of NBS1 as a degenerate site and NBS2 as a functional site within BSEP.

### Disease-causing mutations in the NBDs

Approximately 300 genetic BSEP variants have been identified, which impair or reduce its activity and ultimately lead to cholestasis[28]. The majority of these mutations lack molecular-level characterization due to the absence of structural analysis. Given that our EM studies provided high-resolution experimental structures of the NBDs of BSEP, they allowed us to conduct a comprehensive analysis of eight NBD mutations. Three of them (R432T, Q558H, and R1231Q) are implicated in BRIC2, while the other five (T463I, G562D, A588V, G1116R, and S1120N) are associated with PFIC2 (Fig. 5, Supplementary Fig. 12, and Supplementary Table 3). Our structures of nucleotide-bound BSEP revealed that the BRIC2-causing mutation R432T is located at the interface of the two NBDs, likely disrupting the dimerization of the NBDs during the ATPase cycle (Fig. 5b). The two mutations Q558H and R1231Q are located at the TMD–NBD interface, suggesting that they might interfere with the coupling of ATP hydrolysis to transport (Fig. 5c, d). The five mutations associated with PFIC2 are located at the two NBSs, with distances of <5 Å from the bound nucleotide, indicating potential impairment of ATP binding or hydrolysis (Fig. 5e–i). Specifically, the T463I may lead to a clash with Y429 in NBS1, thereby destabilizing ATP binding. Notably, the function of the mutant T463I can be rescued by ivacaftor, a cystic fibrosis transmembrane conductance regulator (CFTR) potentiator[29]. The mutations G562D and G1116R are expected to directly disrupt ATP binding, whereas A588V and S1120N might impact the binding or conformation of the catalytic glutamate or the Mg$^{2+}$ ions, thereby interfering with ATP hydrolysis. These findings shed light on the functional consequences and the molecular basis of BSEP mutations, providing insight into their pathogenetic mechanism.

### Conformational changes in BSEP

To analyze the conformational changes during substrate (TC) transport and inhibitor (GBM) binding, we compared the movements of the TM helices within the corresponding structures (Fig. 6). In the substrate-free state of BSEP (PDB: 6LR0), the protein adopts an inward-facing conformation, with a distance of ~31 Å between TM3 and TM4 in TMD1. Upon inhibitor binding, the TMDs undergo a subtle closure, resulting in a narrowing of ~2 Å relative to substrate-free BSEP (Fig. 6a). When a molecule of TC binds (PDB: 7E1A), the TM helices are shifted, leading to a relatively narrow opening in the TMDs, as reflected by a reduced distance (~24 Å in TMD1 and ~33 Å in TMD2) compared to the substrate-free state (Fig. 6a). In the presence of the molecules ATP·Mg$^{2+}$·TC, BSEP undergoes substantial conformational changes compared to the previous states. The two sets of helices (TM4 and TM5, TM10 and TM11) are shifted with a reduced distance (~14 Å in TMD1 and ~16 Å in TMD2) toward the other four helices. Additionally, the closure of the central cavity is observed in the structure of BSEP$_{E1244Q}$ and vanadate-trapped BSEP (Fig. 6b). Similar rearrangements of transmembrane helices have been observed in the related transporters ABCB1 and ABCB4[30,31]. Our findings yield insights into the structural basis of conformational

changes on both substrate extrusion and the inhibitory mechanism of BSEP. Furthermore, our structure has shed light on how specific small-molecular inhibitors effectively block the transport process of BSEP.

### Discussion

Our structural data reveal that GBM inhibits BSEP by binding at the apex of the central cavity, preventing TM4 from kinking and the transport cycle from advancing. Other small-molecule inhibitors of BSEP likely bind the same pocket and have similar effects. The mechanism of BSEP inhibition by GBM is reminiscent of that observed for ABCB4 inhibition by posaconazole (Supplementary Fig. 6d), which stabilizes an intermediate substrate-bound state and prevents substrate entry into the pocket by inhibiting TM4 kinking[31]. Despite binding to different regions of the central cavity in their respective transporters, these inhibitors thus appear to employ a similar inhibition mechanism by preventing the transporters from transitioning into an occluded or outward-open state.

Our nucleotide-bound structures of BSEP provide insight into the mechanism of bile salt extrusion. A rearrangement of TM helices may lead to the peristaltic extrusion of the substrate, similar to that postulated for human ABCB1[32]. Three pairs of TM helices (TM3 and TM9, TM4 and TM10, TM6 and TM12) are involved in this process (Supplementary Fig. 13). The substrate-bound structure of BSEP reveals a subtle kinking of TM3 and TM9, TM4 and TM10, resulting in the occlusion of the substrate (Supplementary Fig. 13). Subsequently, as the NBDs close, TM3 and TM9 undergo a transition from the kinked conformation to a straightened conformation (Supplementary Fig. 13). TM4 and TM10 also undergo a conformational change, transitioning from kinking to straightening. TM6 and TM12 move inward towards the vertical pseudo-symmetry axis, resulting in reduced space for substrate binding.

By integrating our findings with previously reported structures of BSEP, we present a mechanistic framework of bile salt transport by BSEP (Supplementary Fig. 14). In the absence of the substrate, the transporter adopts an inward-facing conformation characterized by a central cavity accessible to the cytosol and occupied by the auto-inhibitory helix (state 1). Subsequent binding of the substrate triggers the kinking of TM helices TM3 and TM9, TM4 and TM10, leading to substrate occlusion (state 2). This conformational transition primes the transporter for the subsequent steps. Upon binding of two ATP molecules to the NBSs, the NBDs undergo closure, allowing for BSEP to transition into an outward-open state, which results in substrate release. This is followed by a collapse of the translocation pathway and a rearrangement of the TM helices, making substrate transport essentially irreversible (state 3). Following substrate release, BSEP returns to an inward-facing conformation as the separation of the NBDs is induced by ATP hydrolysis, resetting the transporter for subsequent rounds of transport.

In conclusion, we present the cryo-EM structures of inhibitor-bound and nucleotide-bound BSEP, which may rationalize the small-molecule inhibition of BSEP and visualize BSEP in the collapsed conformation. Our study may help in the design of drugs targeting the liver whilst avoiding inhibition of BSEP and elucidate the structural

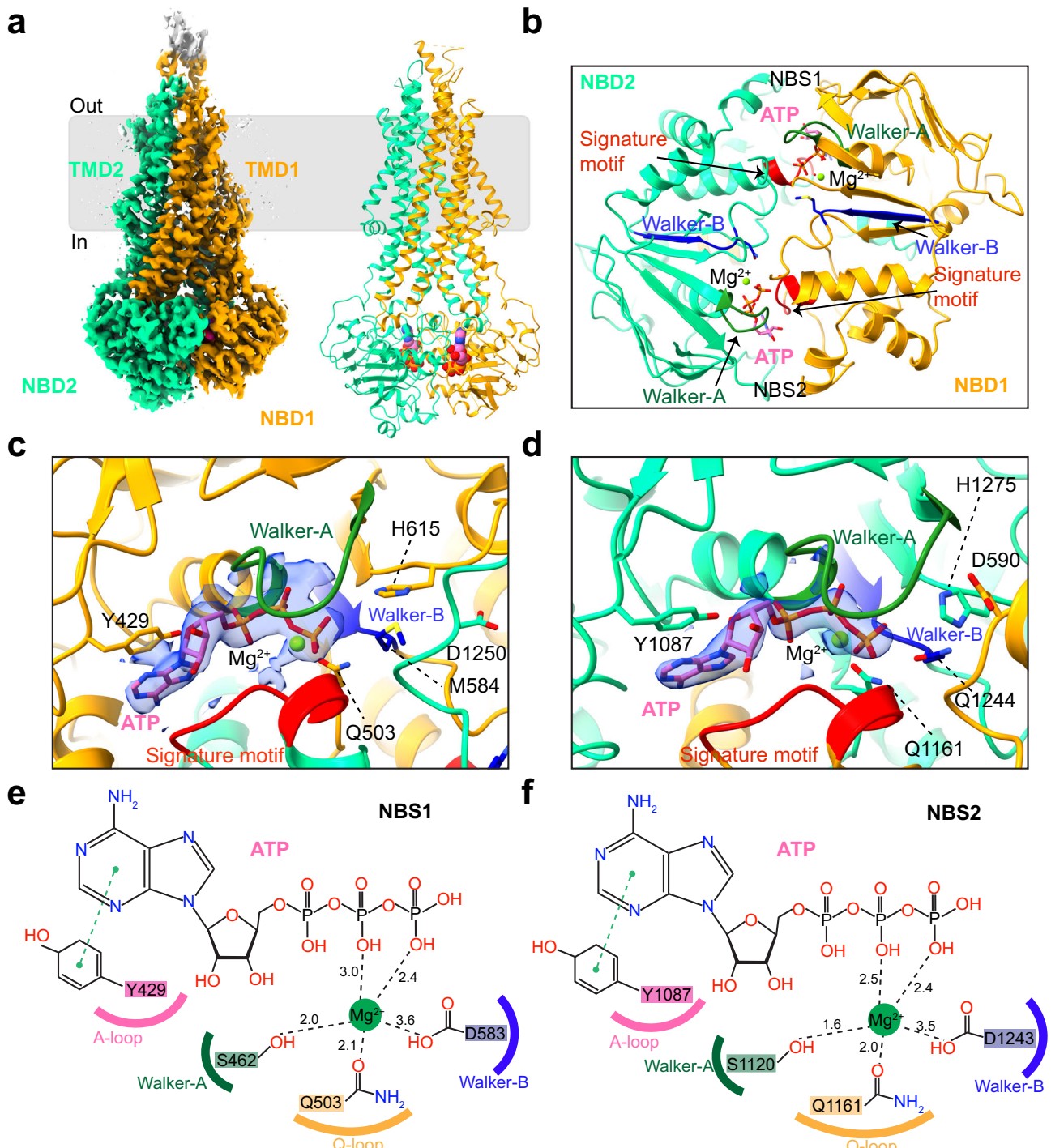

**Fig. 3 | Cryo-EM structure of ATP-bound BSEP_{E1244Q}. a** Cryo-EM density map and model of ATP-bound BSEP_{E1244Q}. The N-terminal half (TMD1 and NBD1) is colored orange and the C-terminal half (TMD2 and NBD2) is green. ATP and Mg$^{2+}$ ions are shown as spheres. **b** Cartoon representation of NBDs viewed from the lipid membrane towards the cytoplasm. **c, d** Molecular interactions of ATP at NBS, shown together with the density of Mg$^{2+}$-ATP (blue density). **e, f** 2D diagrams of the interactions of Mg$^{2+}$-ATP at NBS1 and NBS2, respectively. These conserved motifs (A-loop, Q-loop, Signature motif, Walker-A, and Walker-B) are indicated in (**c–f**).

insights of clinical mutations, especially in NBDs, which may help in the design of new correctors.

## Methods

### Protein expression and purification

The full-length wild-type human BSEP gene (UniProt ID: O95342) was synthesized by GeneArt (Thermo Fisher Scientific) after codon optimization for the mammalian cell expression system. All subsequent modifications of the sequence, including the introduction of point mutations, were performed using synthetic gene fragments and were confirmed by sequencing (Microsynth). Stable cell lines were generated using the Flp-In T-Rex system (Thermo Fisher Scientific) according to the manufacturer's guidelines. The BSEP constructs were fused with a C-terminal eYFP-rho-1D4 tag and a preceding 3C protease cleavage site. Cells were adapted, grown, and maintained in the fresh complete Dulbecco's modified Eagle medium (DMEM, Gibco)

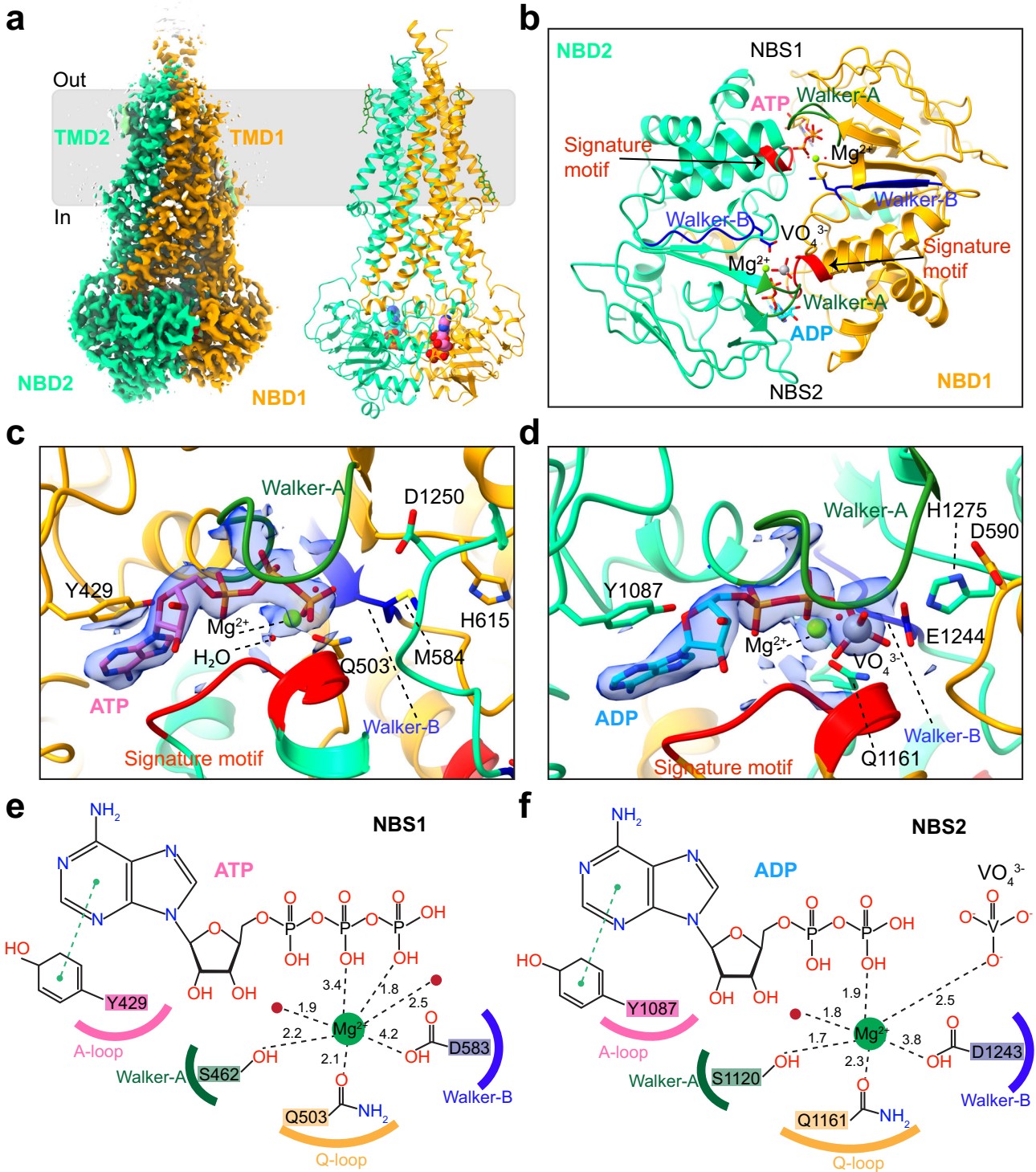

**Fig. 4 | Cryo-EM structure of vanadate-trapped BSEP. a** Cryo-EM density map and model of vanadate-trapped BSEP. The N-terminal half (TMD1 and NBD1) is colored orange and the C-terminal half (TMD2 and NBD2) is colored green. ATP, ADP, and $Mg^{2+}$ ions are shown as spheres. Cholesterols are shown as sticks (dark green). **b** Cartoon representation of NBDs viewed from the lipid membrane towards the cytoplasm. **c** Molecular interactions of ATP at NBS1, shown together with the density of $Mg^{2+}$-ATP (blue density). **d** Molecular interactions of $ADP-VO_4^{3-}$ at NBS2, shown together with the density of $Mg^{2+}$-$ADP$-$VO_4^{3-}$ (blue density). **e, f** 2D diagrams of the interactions of $Mg^{2+}$-ATP or $Mg^{2+}$-$ADP$-$VO_4^{3-}$ at NBS1 and NBS2, respectively.

supplemented with 10% fetal bovine serum (FBS, Thermo Fisher Scientific) at 37 °C under the humidified conditions with 5% $CO_2$.

Wild-type BSEP and $BSEP_{E1244Q}$ expression was induced by adding doxycycline (Sigma) to a final concentration of 3 µg/mL for 48 h under the same conditions. Cells were harvested and flash-frozen in liquid nitrogen for storage at −80 °C. For protein purification, frozen cell

pellets were thawed and homogenized in suspension buffer (10:1 vol/wt) containing 25 mM Hepes (pH 7.5), 150 mM NaCl, 20% glycerol, supplemented with cOmplete EDTA-free protease inhibitor tablets (Roche) and DNase I (Roche), followed by addition of the detergents n-dodecyl-β-D-maltopyranoside (DDM, Anatrace), octaethylene glycol monododecyl ether ($C_{12}E_8$, Anatrace) and cholesteryl hemisuccinate

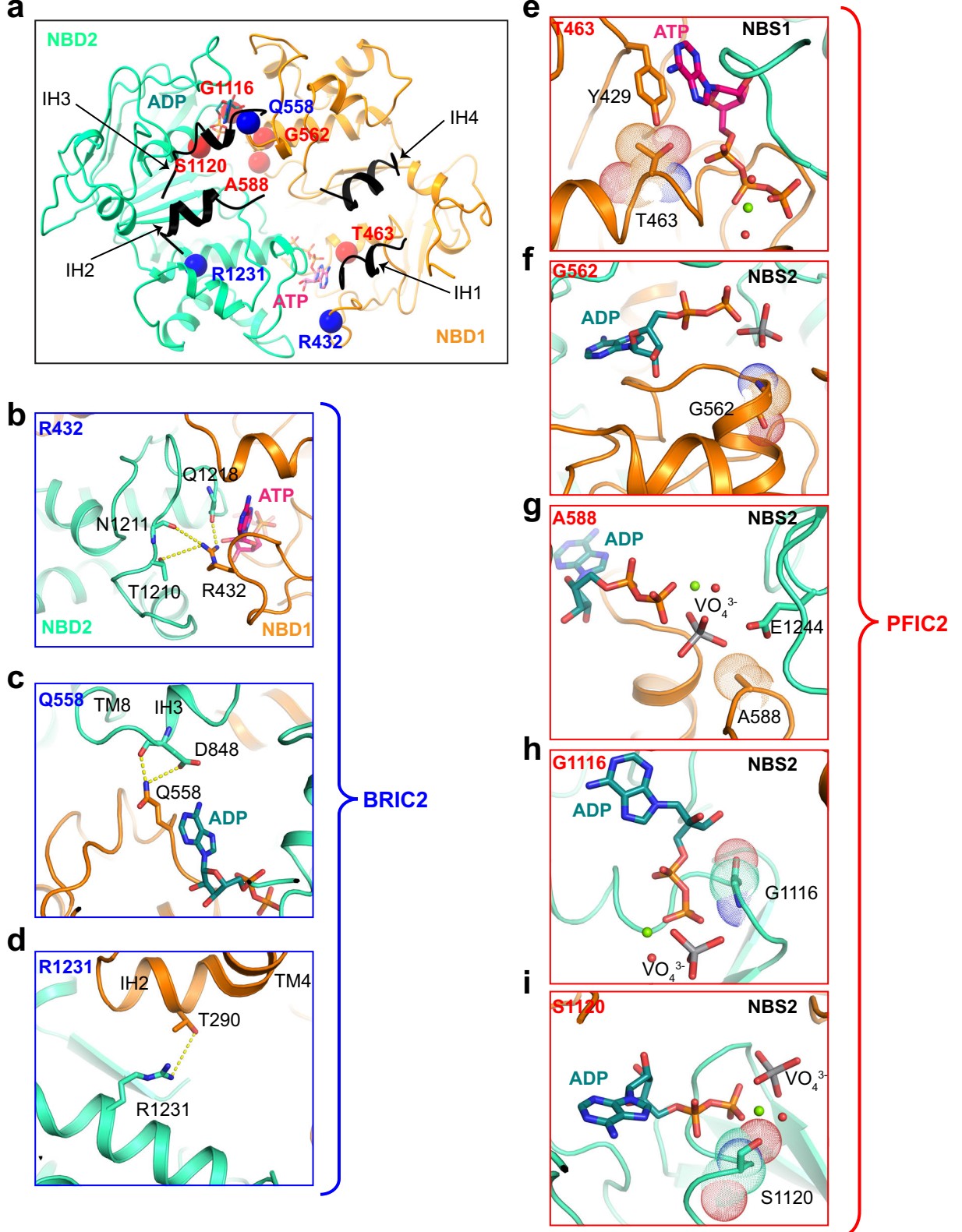

**Fig. 5 | Functionally impairing mutations in the NBDs. a** Mapping of eight clinically relevant mutations in NBDs: R432T, T463I, Q558H, G562D, A588V, G1116R, S1120N, and R1231Q. The red spheres refer to PFIC2 variants, while the blue spheres represent the BRIC2 variants. **b–d** Molecular interactions of BRIC2 variants. The interactions are indicated by dashed lines. **e–i** Molecular interactions of PFIC2 variants. The residues causing PFIC2 are shown as sticks and dots.

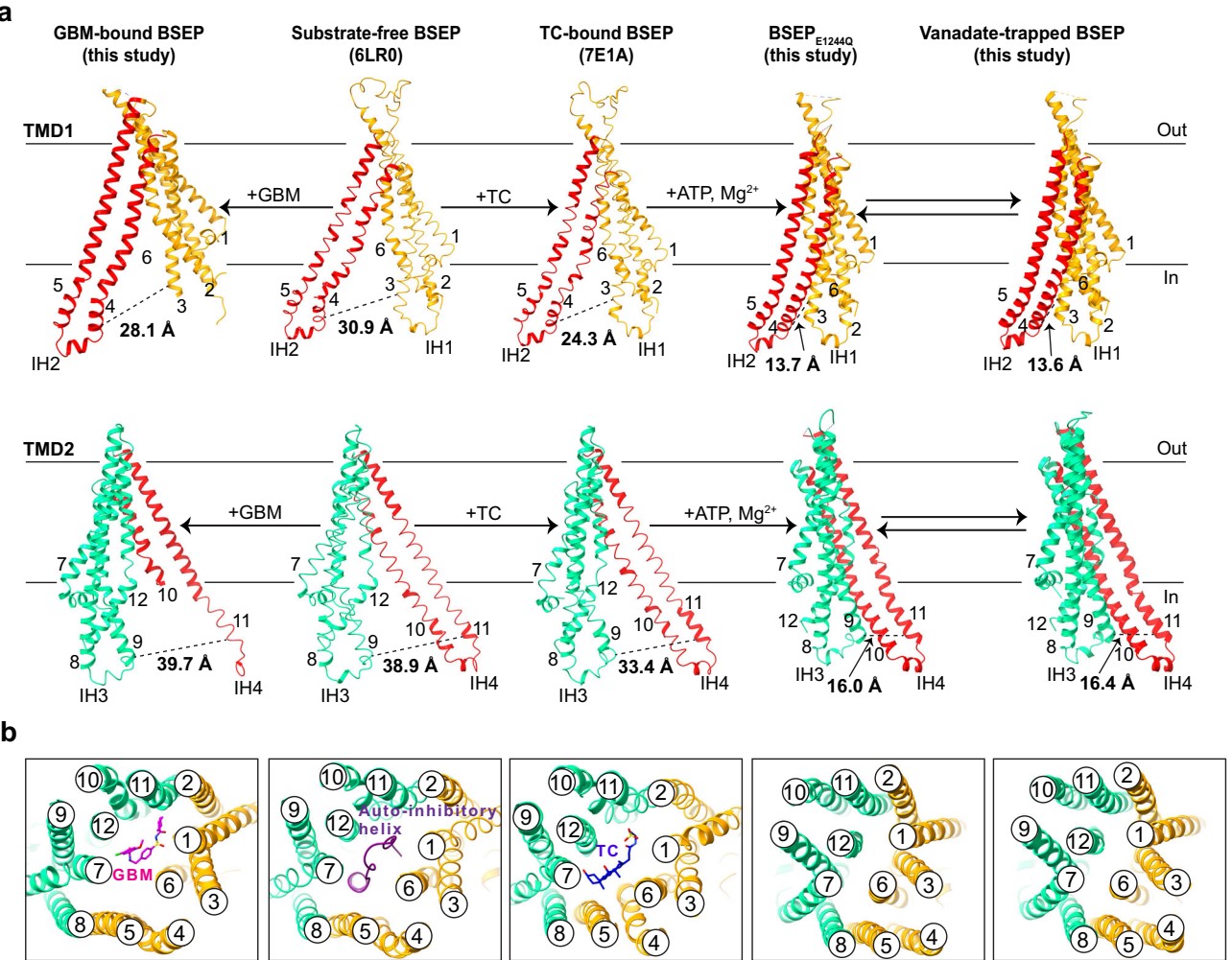

**Fig. 6 | Conformational changes in BSEP. a, b** Cartoon representation of the TMDs in GBM-bound BSEP, substrate-free BSEP, TC-bound BSEP, ATP-bound BSEP$_{E1244Q}$, and vanadate-trapped BSEP. TMDs are numbered and colored orange and green, and TM 4–5 and TM 10–11 are colored red in (**a**). The distances between TM3 (R201) and TM4 (D282) in TMD1 and between TM9 (A856) and TM11 (F959) in TMD2 are measured and labeled. The numbers in the white circles in **b** represent the TM helices.

(CHS, Anatrace) with final concentrations of 0.4% DDM/0.1% $C_{12}E_8$/0.1% CHS (wt/vol). The extraction was performed for 2 h at 4 °C before centrifugation at 140,000 × *g* for 30 min in a Type 45-Ti rotor (Beckmann). The supernatant was applied to an equilibrated Sepharose-coupled rho-1D4 antibody resin (University of British Columbia) and incubated for 2 h at 4 °C. The 1D4 resin was washed three times with 10 column volumes (CVs) of washing buffer containing 25 mM Hepes (pH 7.5), 150 mM NaCl, 20% glycerol, 0.01% DDM, 0.01% $C_{12}E_8$, and 0.004% CHS. Subsequently, the 1D4 resin was incubated for 2 h with three CVs of buffer supplemented with a 1:10 (wt:wt) ratio of 3C protease to protein for cleavage of the C-terminal tags. In addition, reverse Ni-NTA (Qiagen) was carried out to remove the His-tagged 3C protease. The elution was concentrated using a 100 kDa molecular weight cut-off centrifugal filter (Amicon) before being loaded on a Superdex S200 increase column for size exclusion chromatography. The fractions of the peak containing BSEP were collected and the protein concentration in detergent micelles was determined by absorbance at 280 nm using a NanoDrop 2000c spectrophotometer (Thermo Fisher Scientific).

**Nanodiscs reconstitution**

A mixture of brain polar extract lipids (BPL, Avanti Polar Lipids) and cholesterol (Avanti Polar Lipids) (4:1 wt/wt) was solubilized in 1% DDM/

CHS, followed by sonication. The solubilized lipid mixture was mixed with detergent-purified protein and incubated at room temperature for 5 min. Membrane scaffold protein (MSP1D1) was then added to the mixture and incubated for 20 min at room temperature. The stoichiometry of the mixture was maintained at a molar ratio of 1:5:100 (protein to MSP1D1 to lipids). Finally, Bio-Beads SM-2 (Bio-Rad), activated with methanol and pre-equilibrated with HBS buffer (25 mM Hepes pH 7.5, 150 mM NaCl), were added to the nanodiscs mixture at a concentration of 0.8 g/mL and incubated overnight at 4 °C with gentle stirring. The polystyrene beads were removed by passing through a Poly-prep gravity column (Bio-Rad), and then the flow-through mixture was briefly spun down at 4200 × *g* for 5 min at 4 °C to remove excess lipids. The sample was concentrated and purified by size exclusion chromatography (TSKgel G3000SWXL column, 0.35 mL/min) with HBS as the running buffer. Peak fractions containing nanodisc-reconstituted BSEP were collected and the protein concentration was determined using a NanoDrop 2000c spectrophotometer before plunging EM grids.

**Proteoliposomes reconstitution**

Protein reconstitution into proteoliposomes was performed[33]. Briefly, the lipid mixture (BPL and cholesterol) was extruded 11 times through a 400-nm polycarbonate filter (Whatman) and destabilized with 0.25%

(vol/vol) Triton X-100. Detergent-purified BSEP was mixed with the lipid mixture with a 100:1 (wt/wt) lipid: protein ratio, incubated, and reconstituted by adding Bio-Beads SM-2 overnight at 4 °C. The proteoliposomes were centrifuged at 150,000×*g* for 25 min in a Type 70-Ti rotor (Beckmann) and resuspended in HBS buffer to a final lipid concentration of 10 mg/mL. Proteoliposomes containing BSEP were flash-frozen and stored at −80 °C prior to use for activity assays. Proteoliposomes are extruded 11 times through a 400-nm polycarbonate filter before the assay to make them unilamellar. To measure the reconstitution efficiency of BSEP and the concentration of BSEP in the proteoliposomes, the Schaffner and Weissmann protein assay was used[34].

## ATPase activity assay

ATP hydrolysis was measured using a molybdate-based colorimetric assay[35]. The assay was conducted to measure the amount of inorganic phosphate released from the BSEP-mediated ATP hydrolysis reactions under different conditions. The BSEP sample was used at ~5 μg of protein (detergent) or ~2.5 μg of protein (proteoliposomes) per reaction. The ATPase assays were initiated by the addition of 5 mM ATP in the presence of 10 mM MgCl₂, incubated for 0–30 min at 37 °C, and stopped by the addition of 6% SDS. Basal activity reactions in the absence of substrate were performed. All reactions were run in triplicate and repeated with different batches of protein preparation and reagents. Results were analyzed, and curves were plotted using the non-linear regression Michaelis-Menten analysis tool in GraphPad Prism 9.

## Transport assay

The uptake of radioactive taurocholate (³H-TC, American Radiolabeled Chemicals) was used to determine the transport activity of BSEP. All reactions were performed in the HBS buffer at room temperature. 0.5 μM ³H-TC with 50 μM unlabeled TC (Sigma) was added and the sample was incubated for 5 min at room temperature. The transport reaction was initiated by the addition of 5 mM ATP and 10 mM MgCl₂. To stop the reaction at each time point (0, 1, 2, 5, 10, 20, 30, 45 min), samples were removed and then filtered through a Multiscreen vacuum manifold (MSFBN6B filter plate, Millipore) and washed twice with cold HBS buffer. Radioactivity trapped on the filters was measured using a Perkin Elmer 2450 Microbeta2 microplate scintillation counter. Data were analyzed and curves were plotted using the non-linear regression Michaelis–Menten analysis tool in GraphPad Prism 9.

## Cryo-EM sample preparation

Nanodisc-reconstituted BSEP or BSEP_{E1244Q}, at ~0.8 mg/mL, was equilibrated with either 5 mM ATP, 5 mM MgCl₂, 50 μM TC, 1 mM Na₃VO₄ (vanadate-trapped BSEP) or 5 mM ATP, 5 mM MgCl₂, and 50 μM TC (BSEP_{E1244Q}) or 100 μM GBM (GBM-bound BSEP) for 5 min at room temperature. Samples (3.5 μL) were applied to glow-discharged Quantifoil R1.2/1.3 carbon/copper 300 mesh grids, and then the grids were plunge-frozen in a liquid ethane–propane mixture cooled by liquid nitrogen using a Vitrobot Mark IV (FEI) at 4 °C and 100% humidity.

## Cryo-EM data acquisition and processing

The grids of GBM-bound BSEP were screened on a Glacios TEM (Thermo Fisher Scientific, TFS 200 kV) to determine ice thickness and quality and then transferred to a Titan Krios G4 TEM (TFS 300 kV) equipped with a Cold-FEG, SelectrisX energy filter, and Falcon IV detector. Semi-automated data collection was performed using EPU v2.12.1 (TFS) software. Movies were recorded at a nominal magnification of 165,000x, corresponding to a physical pixel size of 0.726 Å/pix. The defocus at exposure ranged from −0.8 to −2.5 μm, with a calibrated total dose of 60 e⁻/Å².

The BSEP_{E1244Q} and vanadate-trapped BSEP samples were imaged on a Titan Krios 300 kV (TFS) equipped with a Gatan Biocontinuum energy filter and a Gatan K3 detector. Data were acquired with EPU

(Thermo Fisher Scientific) at a nominal magnification of 130,000x and in super-resolution mode (0.33 Å/pix). The defocus ranged from −0.6 to −2.4 μm, with a total dose of 48 and 66 e⁻/Å², respectively. Data acquisition statistics are presented in Supplementary Table 1. The data processing pipelines of GBM-bound BSEP, BSEP_{E1244Q}, and vanadate-trapped BSEP are presented in the Supplementary Figures.

The multi-frame movies of GBM-bound BSEP were imported into Relion 4.0[36] and were motion-corrected (MotionCor2)[37], dose-weighted, and binned twice to a pixel size of 0.726 Å/pix. The CTF was estimated by CTFFIND4[38]. We auto-picked 5,180,029 particles using the Laplacian-of-Gaussian method, followed by particle extraction with a pixel size of 2.178 Å/pix. After several rounds of 2D classification, 953,008 particles were selected for 3D classification. The initial model was generated in CryoSPARC v3.3. The particles from the best resolved 3D class were re-extracted with an unbinned pixel size of 0.726 Å/pix, 3D auto-refined, and 3D classified without alignment and with mask, yielding a map at 3.48 Å. The 3D auto-refinement was followed by Bayesian Polishing, CTF refinement, further 3D refinement, and post-processing. The final map used for model building was at a resolution of 3.22 Å. The local resolution map was calculated in Relion 4.0[36].

The multi-frame movies of BSEP_{E1244Q} and vanadate-trapped BSEP samples were imported into Relion 3.1[39] and were motion-corrected (MotionCor2)[37], dose weighted and binned twice to a pixel size of 0.66 Å/pix. The contrast transfer function (CTF) was estimated by Gctf[40]. For the BSEP_{E1244Q} sample, 5,250,101 particles were auto-picked based on a previously generated 2D template from 13,631 micrographs, followed by particle extraction with a 4-fold binned pixel size of 2.64 Å/pix. An initial model was generated from the best 2D classes. After 3 rounds of 2D classification, 1,315,094 particles were subjected to 3D classification. The best classes were selected and re-extracted with a binned pixel size of 1.32 Å/pix. After several rounds of 3D classification and refinement, we obtained an EM density map at a resolution of 3.33 Å. The particles were re-extracted with an un-binned pixel size of 0.66 Å/pix. The data was then subjected to 3D classification and per-particle CTF Refinement, followed by another 3D refinement with a mask excluding the nanodiscs, resulting in a 2.95 Å resolution EM density map.

For the vanadate-trapped BSEP sample, we auto-picked 5,886,181 particles from a total of 18,554 micrographs using the Laplacian-of-Gaussian method, followed by particles extraction with a 4-fold binned pixel size of 2.64 Å/pix. After several rounds of 2D classification, 1,779,457 particles were selected for 3D classification. Here, the best classes were selected to generate an initial model, and 3D classification was performed. The particles from the best resolved 3D classes were re-extracted with an un-binned pixel size of 0.66 Å/pix, 3D auto-refined, and 3D classified without alignment and with a mask. Next, the data were subjected to Bayesian Polishing, per-particle CTF refinement, and another 3D refinement with the mask excluding the nanodiscs, resulting in a final EM density map at 2.81 Å resolution. Local resolution was calculated in Relion 3.1[39].

## Model building and refinement

The final postprocessed cryo-EM maps were used for model building in Coot[41]. The cryo-EM map of GBM-bound BSEP showed good EM densities in TMDs and NBD2, except for NBD1. We built the model of GBM-bound BSEP based on the previously reported structure of substrate-free BSEP (PDB ID: 6LR0)[22]. The N-terminus (residues 1–43), extracellular region (residues 103–128), intracellular region (residues 188–197), NBD1 region with the linker connecting TMD2 (residues 399–735), and half of TM10 (residues 925–950) are not built due to poorly resolved density. The two EM maps of BSEP_{E1244Q} and vanadate-trapped BSEP have nicely resolved TMDs and NBDs, allowing for de novo model building based on the protein sequence. In the final model of vanadate-trapped BSEP, the N-terminus (residues 1–43), extracellular region (residues 101–127, 346–354, and 783–788), and

intracellular connection (residues 638–736) are not built because of the weaker density. As for BSEP$_{E1244Q}$, these regions are also not built (residues 1–43, 101–127, 347–352, 638–738, 782–789, and 1011–1012). Model refinements were performed in Phenix[42] with geometric and secondary structure restraints.

## Figure preparation
Graph preparation and statistical analysis were performed in GraphPad 9 (macOS, GraphPad Software, La Jolla, CA, USA, www.graphpad.com). All the images of models and EM maps were prepared in PyMOL (the PyMOL Molecular Graphics System, Version 2.5, Schrödinger), UCSF Chimera[43], and UCSF Chimera X[44].

## Reporting summary
Further information on research design is available in the Nature Portfolio Reporting Summary linked to this article.

## Data availability
The data that support this study are available from the corresponding authors upon request. The atomic coordinates of ATP-bound BSEP$_{E1244Q}$, vanadate-trapped BSEP, and GBM-bound BSEP models from this study have been deposited in the Protein Data Bank (PDB) under accession codes 8PMD (ATP-bound BSEP$_{E1244Q}$), 8PMJ (vanadate-trapped BSEP), and 8PM6 (GBM-bound BSEP). The cryo-EM maps have been deposited in the Electron Microscopy Data Bank (EMDB) under accession codes EMD-17759 (ATP-bound BSEP$_{E1244Q}$), EMD-17761 (vanadate-trapped BSEP), and EMD-17758 (GBM-bound BSEP). The source data underlying Figs. 1b–d, 2a–f, and Supplementary Figs. 1–3 are provided as a Source Data file. Source data are provided with this paper.

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

## Acknowledgements

We thank the Scientific Center for Optical and Electron Microscopy (ScopeM) facility at ETH Zürich and the Dubochet Center for Imaging at EPFL/University of Lausanne for support with cryo-EM data collection. This research was supported by Swiss National Science Foundation grant 310030_189111 (to K.P.L.), grant IZLCZO_206089 (to H.S.), and the NCCR Transcure (grant 185544).

## Author contributions

H.L., R.N.I., and K.P.L. conceived the project. H.L. expressed and purified human BSEP. H.L., R.B.-S., and L.L. conducted functional assays. R.N.I. and D.N. collected cryo-EM data. H.L., R.N.I., J.K., and K.N. processed and analyzed cryo-EM data and built the atomic models. B. S. and H. S. helped analyze functional data and cryo-EM data, respectively. All authors analyzed data. H.L., J.K., and K.P.L. wrote the manuscript with input from R.N.I., R.B.-S., B.S., D.N., and H.S.

## Competing interests

The authors declare no competing interests.
