## [Peer Review File · Nature Communications]

Structural basis of bile salt extrusion and small-molecule inhibition in human BSEPReviewers' Comments:

Reviewer #1:

Remarks to the Author:

In the manuscript "Structural basis of bile salt extrusion and small-molecule inhibition in human BSEP" Liu, Irobalieva, et al. have undertaken extensive structural studies, combined with functional studies, to describe the mechanism and inhibition of the human bile salt export pump (BSEP), an ABC transporter of importance in bile salt extrusion in the liver. The molecular mechanism and small molecule inhibition have not been understood to this point, despite of available structures of BSEP solubilized in detergent. BSEP is of particular importance in mediating the efflux of bile salts, which are key components of bile. Mutations and common drugs, in particular glibenclamide/glyburide used in the treatment of type 2 diabetes, are responsible for severe cholestatic liver disease and damage. Thus, the structures and mechanism may very well be the key to developing new strategies for modulation of the BSEP function, which will be of significant interest to readers from a wide range of fields.

The strategy to reconstitute BSEP in nanodiscs for structural studies was chosen based on the authors' activity measurements of BSEP reconstituted in proteoliposomes vs. in detergent, which showed 8-fold higher activity in proteoliposomes. Furthermore, they demonstrated that the small molecule inhibitor glibenclamide (GBM) stimulates BSEP activity 2-fold. Additional functional studies revealed that taurocholate transport is dependent on ATP, and the transport rate and turnover were established. Vanadate inhibits ATPase activity and taurocholate (TC) transport, while GBM decreases ATPase activity and taurocholate uptake.

In order to gain an understanding of the mechanism and inhibition, cryo-EM structural studies were performed on BSEP reconstituted in nanodiscs, in particular with the small molecule inhibitor glibenclamide (GBM) bound, of pre- and posthydrolysis BSEP structures with ATP bound, and of an E1244Q mutant, for which the authors could not detect activity.

The structure with GBM revealed that binding of GBM occurs deeper in the binding pocket than TC binding, which prevents a conformational change. Subsequently, active site residues were mutated to test their significance via ATPase activity measurements. The vanadate-bound BSEP at 2.8Å resolution shows the trapping of ADP. Both this structure and the mutant structure appear to show states after substrate release. The mutant shows the state before hydrolysis, while the vandate-trapped BSEP structure shows the state before the release of hydrolyzed nucleotides. Combined with the structures of nucleotide-, substrate- and inhibitor-bound BSEP, as well as the well-resolved densities for ATP and Mg+2, the structural and functional data allowed for a detailed description of conformational changes and the mechanism. Furthermore, structural studies of 8 NBD mutations resulted in a description of the detailed consequences of each mutant on different aspects of function, which will be of immense significance in considering strategies to target the associated diseases.

It is not an understatement by the authors that their rigorous characterization of the active site and mechanism may have substantial consequences for drug design that targets the liver while preventing binding to and inhibition of BSEP. Thus, publication is strongly recommended.

Minor comments:

p. 3, line 57: The abbreviation for "TC" is either missing, or not clear in this context.

p. 5, line 102: The authors may want to consider moving Figures 2d-f to Figure 3 to follow the order of discussion.

p. 5, line 105-106: Please clarify why two distinct GBM orientations appear possible. At this point in the text, it may be helpful to mention that experiments to test the orientation are described later in

this section.

p. 7, line 162: The first two sentences in the paragraph should be rewritten for clarity.

p. 10, line 239: Change "allowing to" to "allowing for"

p. 14, lines 346 and 351: Which of the two different nominal magnifications was used?

p. 15: Two different versions of Relion were used. An explanation of the reason would be helpful, even if the reason was that the two versions were installed on different machines.

p. 17, lines 422-425: Not all of the authors are mentioned under the contributions.

Supplementary Fig. 14: The shade of blue ADP is difficult to discern on the green background. Please consider using a darker blue or another color.

Reviewer #2:

Remarks to the Author:

The human ABC transporter BSEP or ABCB11 is obviously an important hepatobiliary transporter of the liver. Together with ABCB4 and ABCG5/G8, ABCB11 transports the major components of bile, PC, cholesterol, and bile salts. It is therefore not of great surprise that mutations in ABCB11 are responsible for liver dysfunction and / or failure.

In the submitted manuscript, the Locher group reports three single particle cryo-EM structures of human BSEP reconstituted in nanodiscs. These structures are glibenclamide bound ABCB11 in the inward facing conformation and ABCB11 with bound ATP or trapped with ADP*Vi. In the latter one, the degenerated NBS, NBS1, contains a bound ATP molecule, while the active NBS, NBS2, contains ADP and Vi. For the ATP-bound structure an ATPase deficient mutant, the EQ mutant, was employed. Major conclusions derived from the obtained structures are validated by mutational studies. These three, new structures together with the published structures of ABCB11, apo-ABCB11 in the inward facing conformation and ABCB11 with one or two bound taurocholate (TC) molecules also in the inward facing conformation lay the foundation for the formulation of a transport cycle of ABCB11.

In light of the importance of ABCB11, these three structures are important and might deserve publication in Nature Communications. However, a couple of points have to be clarified prior to a final decision. The order of my points is not ranked according to importance but rather to appearance in the text:

- Glibenclamide is known as a competitive inhibitor of ABCB11. This is also in agreement with the position observed in the structure. However, I don't grasp why the ATPase activity of ABCB11 reconstituted in liposomes (Figure 1c) is nearly twice as high as the basal activity in the presence of 0.5 mM glibenclamide. Especially in the light of the statement of the authors (line 113 on page 5) that "GBM thus locks the transporter in an unproductive, inward-facing conformation, preventing it from undergoing conformational changes". If the transporter is locked then ATPase activity is not present as the conformational changes are a prerequisite for activity.

- How does the transporter behave in detergent solution if activity is analyzed in the presence of glibenclamide?

- Figure 1c: the fact that vanadate trapping in the presence of 0.5 mM taurocholate reduces activity is also odd. Simply stating that the reason is "unclear" is not justified as it also might indicate that TC interferes with vanadate trapping. What about other well-known ATPase inhibitors such as beryllium or aluminum fluoride?

- Based on the inhibitor-bound structure, interacting residue side chains were identified. However, a

mutational analysis identified only one out of six mutations that abolished glibenclamide stimulated ATPase activity. Two even stimulated ATPase activity further than the wild type protein! Here I like to point again to the author's statement on page 5 (line 111). How is ATPase stimulation possible if glibenclamide locks the transporter? More important, the results of the mutational studies point to a situation in which only one out of six residues is required for binding. This in my opinion is very odd. Does it mean that the observed binding pose of glibenclamide is artificial and does not reflect the physiological situation? I would request to see combination of these mutations and that residue 80 is not only replaced by F but also other residues such as V or A.

- Figure S3: why is no signal for empty nanodiscs observed?

- Figure 3C: I have to admit that the density for one of the two water molecules is not convincing, while no water molecules have been fitted in the density of Figure S8. Here, the density is much more convincing.

Reviewer responses

We thank the reviewers for their encouraging and insightful comments. We have made several changes in the manuscript to address the concerns raised. The reviewers' comments (with numbers indicated) are in black and our responses are colored blue for clarity.

Reviewer #1 (Remarks to the Author):

In the manuscript "Structural basis of bile salt extrusion and small-molecule inhibition in human BSEP" Liu, Irobalieva, et al. have undertaken extensive structural studies, combined with functional studies, to describe the mechanism and inhibition of the human bile salt export pump (BSEP), an ABC transporter of importance in bile salt extrusion in the liver. The molecular mechanism and small molecule inhibition have not been understood to this point, despite of available structures of BSEP solubilized in detergent. BSEP is of particular importance in mediating the efflux of bile salts, which are key components of bile. Mutations and common drugs, in particular glibenclamide/glyburide used in the treatment of type 2 diabetes, are responsible for severe cholestatic liver disease and damage. Thus, the structures and mechanism may very well be the key to developing new strategies for modulation of the BSEP function, which will be of significant interest to readers from a wide range of fields.

The strategy to reconstitute BSEP in nanodiscs for structural studies was chosen based on the authors' activity measurements of BSEP reconstituted in proteoliposomes vs. in detergent, which showed 8-fold higher activity in proteoliposomes. Furthermore, they demonstrated that the small molecule inhibitor glibenclamide (GBM) stimulates BSEP activity 2-fold. Additional functional studies revealed that taurocholate transport is dependent on ATP, and the transport rate and turnover were established. Vanadate inhibits ATPase activity and taurocholate (TC) transport, while GBM decreases ATPase activity and taurocholate uptake.

In order to gain an understanding of the mechanism and inhibition, cryo-EM structural studies were performed on BSEP reconstituted in nanodiscs, in particular with the small molecule inhibitor glibenclamide (GBM) bound, of pre- and posthydrolysis BSEP structures with ATP bound, and of an E1244Q mutant, for which the authors could not detect activity.

The structure with GBM revealed that binding of GBM occurs deeper in the binding pocket than TC binding, which prevents a conformational change. Subsequently, active site residues were mutated to test their significance via ATPase activity measurements. The vanadate-bound BSEP at 2.8Å resolution shows the trapping of ADP. Both this structure and the mutant structure appear to show states after substrate release. The mutant shows the state before hydrolysis, while the vandate-trapped BSEP structure shows the state before the release of hydrolyzed nucleotides. Combined with the structures of nucleotide-, substrate- and inhibitor-bound BSEP, as well as the well-resolved densities for ATP and Mg²⁺, the structural and functional data allowed for a detailed description of conformational changes and the mechanism. Furthermore, structural studies of 8 NBD mutations resulted in a description of the detailed consequences of each mutant on different aspects of function, which will be of immense significance in considering strategies to target the associated diseases.

It is not an understatement by the authors that their rigorous characterization of the active site and mechanism may have substantial consequences for drug design that targets the liver while preventing binding to and inhibition of BSEP. Thus, publication is strongly recommended.

We thank the reviewer for this positive evaluation.

Minor comments:

1.1 p. 3, line 57: The abbreviation for "TC" is either missing, or not clear in this context.

Corrected in the revised manuscript.

1.2 p. 5, line 102: The authors may want to consider moving Figures 2d-f to Figure 3 to follow the order of discussion.

We are unsure why the reviewer suggests moving panels d-f of Figure 2 to Figure 3. We would like to clarify that the two figures show two distinct cryo-EM structures. Figure 2 shows the structure of inhibitor-bound BSEP, whereas Figure 3 shows that of ATP-bound BSEP_{E1244Q}. We feel it would be confusing for the reader to mix these figures.

1.3 p. 5, line 105-106: Please clarify why two distinct GBM orientations appear possible. At this point in the text, it may be helpful to mention that experiments to test the orientation are described later in this section.

The structure of GBM can be thought to have pseudo-2-fold symmetry at first approximation (and at low resolution). This provides two possible ways to fit GBM into the cryo-EM density. We think our structure captures a combination of the two binding poses, because the data processing algorithm cannot distinguish between the two possibilities. However, as indicated in Fig. 2c and d, the side chain of residue F1018 is likely to adopt two distinct rotamers depending on the binding pose/orientations of GBM within the binding pocket.

1.4 p. 7, line 162: The first two sentences in the paragraph should be rewritten for clarity.

We have rewritten the first sentence as follows: "The structures of BSEP_{E1244Q} and vanadate-trapped BSEP reveal a conformation of the TMDs with a closed (collapsed) translocation pathway, with no space for a substrate or inhibitor. In the context of a transport cycle, these structures therefore likely reflect states after substrate release but prior to the release of hydrolyzed nucleotides from the NBDs (Figs. 3a, 4a)."

1.5 p. 10, line 239: Change "allowing to" to "allowing for"

Corrected.

1.6 p. 14, lines 346 and 351: Which of the two different nominal magnifications was used?

As indicated in the Methods section, a magnification of 165,000 was used for the inhibitor-bound BSEP structure, and a magnification of 130,000x was used for the nucleotide-bound samples. The reason for the difference is that the data were collected using two different Titan Krios cryo-TEMs.

1.7 p. 15: Two different versions of Relion were used. An explanation of the reason would be helpful, even if the reason was that the two versions were installed on different machines.

Distinct versions of Relion were installed on different processing computers used for data processing.

1.8 p. 17, lines 422-425: Not all of the authors are mentioned under the contributions.

We have updated the section on author contributions.

1.9 Supplementary Fig. 14: The shade of blue ADP is difficult to discern on the green background. Please consider using a darker blue or another color.

We changed the color to a darker shade.

Reviewer #2 (Remarks to the Author):

The human ABC transporter BSEP or ABCB11 is obviously an important hepatobiliary transporter of the liver. Together with ABCB4 and ABCG5/G8, ABCB11 transports the major components of bile, PC, cholesterol, and bile salts. It is therefore not of great surprise that mutations in ABCB11 are responsible for liver dysfunction and / or failure.

In the submitted manuscript, the Locher group reports three single particle cryo-EM structures of human BSEP reconstituted in nanodiscs. These structure are glibenclamide bound ABCB11 in the inward facing conformation and ABCB11 with bound ATP or trapped with ADP*Vi. In the latter one, the degenerated NBS, NBS1, contains a bound ATP molecule, while the active NBS, NBS2, contains ADP and Vi. For the ATP-bound structure an ATPase deficient mutant, the EQ mutant, was employed. Major conclusions derived from the obtained structures are validated by mutational studies. These three, new structures together with the published structures of ABCB11, apo-ABCB11 in the inward facing conformation and ABCB11 with one or two bound taurocholate (TC) molecules also in the inward facing conformation lay the foundation for the formulation of a transport cycle of ABCB11.

In light of the importance of ABCB11, these three structures are important and might deserve publication in Nature Communications. However, a couple of points have to be clarified prior to a final decision. The order of my points is not ranked according to importance but rather to appearance in the text:

We thank the reviewer for this positive evaluation.

2.1 - Glibenclamide is known as a competitive inhibitor of ABCB11. This is also in agreement with the position observed in the structure. However, I don't grasp why the ATPase activity of ABCB11 reconstituted in liposomes (Figure 1c) is nearly twice as high as the basal activity in the presence of 0.5 mM glibenclamide. Especially in the light of the statement of the authors (line 113 on page 5) that "GBM thus locks the transporter in an unproductive, inward-facing conformation, preventing it from undergoing conformational changes". If the transporter is locked than ATPase activity is not present as the conformational changes are a prerequisite for activity.

We agree that our choice of words was confusing. We have revised the sentence for clarity (page 5): "Thus, GBM promotes an unproductive, inward-facing conformation, thereby interfering with conformational changes required for substrate extrusion, but not with those associated with ATP hydrolysis, as reflected by the stimulation of the ATPase activity in the presence of GBM (Figs. 1c, 2a)."

It is important to note that glibenclamide indeed stimulates BSEP ATPase activity (2-fold in proteoliposomes), consistent with observations in earlier studies¹. Similar stimulation by an inhibitor was also observed with the multidrug transporter ABCB1, the ATPase activity of which is stimulated by verapamil and other transport inhibitors². Thus, stimulation of ATPase activity by transport inhibitors is not uncommon in the B family of ABC transporters.

1. Noe. J., Hagenbuch. B., Meier. P. J., St-Pierre. M. V. Characterization of the mouse bile salt export pump overexpressed in the baculovirus system. *Hepatology*. **33** (5), 1223-31 (2001)
2. Nosol. K., Romane. K., Irobalieva. R. N., Alam. A., Kowal. J., Fujita. N., Locher. K. P. Cryo-EM structures reveal distinct mechanisms of inhibition of the human multidrug transporter ABCB1. *Proc Natl Acad Sci U S A*. **117** (42), 26245-26253 (2020)

2.2 - How does the transporter behave in detergent solution if activity is analyzed in the presence of glibenclamide?

Figure 2a shows the ATPase activity of BSEP in detergent with TC and glibenclamide. Our results demonstrate that the ATPase activity of BSEP increases in the presence of glibenclamide (but not as high as when stimulated with the substrate TC).

2.3 - Figure 1c: they fact that vanadate trapping In the presence of 0.5 mM taurocholate reduces activity is also odd. Simply stating that the reason is "unclear" is not justified as it also might indicate that TC interferes with vanadate trapping. What about other well-known ATPase inhibitors such as beryllium or aluminum fluoride?

We are unsure what the reviewer refers to. We used the term "unclear" only once, but not in the context of trapping. We state that the precise reason why the basal ATPase activity of certain BSEP mutants is increased compared to wild type is unclear. With respect to trapping, vanadate trapping is a well-known method for studying the B-family of ABC transporters, particularly in the presence of substrate. Kim *et al* determined the cryo-EM structure of human ABCB1 with EQ mutation in nucleotide-binding sites in the presence of substrate³. Mi *et al* determined the cryo-EM structures of the bacterial ABC transporter Msb A in the presence of substrate, ATP/ADP, Mg²⁺, and sodium vanadate⁴. It is important to note that the two ATPase inhibitors mentioned by the reviewer, beryllium and aluminum fluoride, do not inherently offer advantages over sodium vanadate for structural trapping of nucleotide-bound states.

3. Kim. Y., Chen. J. Molecular structure of human P-glycoprotein in the ATP-bound, outward-facing conformation. *Science*. **359**, 915-919 (2018)
4. Mi. W., Li. Y., Yoon. S. H., Ernst. R. K., Walz. T., Liao. M. Structural basis of MsbA-mediated lipopolysaccharide transport. *Nature*. **549**, 233-237 (2017)

2.4 - Based on the inhibitor-bound structure, interacting residue side chains were identified. However, a mutational analysis identified only one out of six mutations that abolished glibenclamide stimulated

ATPase activity. Two even stimulated ATPase activity further than the wild type protein! Here I like to point again to the author's statement on page 5 (line 111). How is ATPase stimulation possible if glibenclamide locks the transporter? More important, the results of the mutational studies point to a situation in which only one out of six residues is required for binding. This in my opinion is very odd. Does it mean that the observed binding pose of glibenclamide is artificial and does not reflect the physiological situation? I would request to see combination of these mutations and that residue 80 is not only replaced by F but also other residues such as V or A.

For these mutants, we analyzed the basal ATPase activity vs. the ATPase activity in the presence of GBM. The results were interpreted based on the fold of stimulation of each mutant compared to the basal activity. We demonstrate that WT BSEP shows the greatest stimulation, with a 3.6-fold increase. All other mutants show lower levels of stimulation. We interpret this as decreased contacts of the binding surface of BSEP with GBM. It is noteworthy that almost all mutants analyzed appear to maintain some interaction with glibenclamide. The exception is L80F, which shows no stimulation by GBM. We conclude that this is likely due to the increased size of the side chain (phenyl ring of phenylalanine), which most likely causes a steric clash with GBM and therefore prevents it from binding.

A combination of these mutations is likely to have a similar effect as already demonstrated, perhaps resulting in a further decrease of GBM-stimulated ATPase activity. We expect that any mutations that cause a steric clash with GBM would show no GBM-stimulated ATPase activity. Given that generating additional double mutants of BSEP constitutes a very significant amount of extra work for very limited additional insight, we feel that further mutations are unwarranted in the context of this manuscript and will unduly delay the publication of our results.

To clarify the insight gained from our mutational study, we have modified Fig. 2f by adding the "fold stimulation" of the ATPase activity. We have also added the statistical significance using Tukey's test in Figs. 1b,c,d and Fig. 2f.

2.5 - Figure S3: why is not signal for empty nanodiscs observed?

We optimized the protocol of nanodiscs formation with a molar ratio of protein:MSP:lipids of 1:5:100, as stated in the manuscript. Under these conditions, only a small fraction of empty nanodiscs is observed, as shown in Supplementary Fig. 3. We included a label in the panel to denote the corresponding fraction.

2.6 - Figure 3C: I have to admit that the density for one of the two water molecules is not convincing, while no water molecules have been fitted in the density of Figure S8. Here, the density is much more convincing.

We agree and have removed the water molecules from Fig. 3 for clarity.

Reviewers' Comments:

Reviewer #1:

Remarks to the Author:

The reviewer comments for manuscript NCOMMS-23-32761A entitled " Structural basis of bile salt extrusion and small-molecule inhibition in human BSEP were carefully considered and adequately addressed.

Reviewer #2:

Remarks to the Author:

All my points have been adequately addressed and I recommend acceptance.

Reviewers' Comments:

Reviewer #1 (Remarks to the Author):

The reviewer comments for manuscript NCOMMS-23-32761A entitled " Structural basis of bile salt extrusion and small-molecule inhibition in human BSEP were carefully considered and adequately addressed.

Reviewer #2 (Remarks to the Author):

All my points have been adequately addressed and I recommend acceptance.

We thank the reviewers for their positive evaluation.